# Genotypic Characteristics and Correlation of Epidemiology of *Staphylococcus aureus* in Healthy Pigs, Diseased Pigs, and Environment

**DOI:** 10.3390/antibiotics9120839

**Published:** 2020-11-24

**Authors:** Yuanyuan Zhou, Xinhui Li, He Yan

**Affiliations:** 1School of Food Science and Engineering, South China University of Technology, Guangzhou 510000, China; 201821025708@mail.scut.edu.cn; 2Department of Microbiology, University of Wisconsin-La Crosse, 1725 State Street, La Crosse, WI 54601, USA; xli@uwlax.edu; 3Research Institute for Food Nutrition and Human Health, Guangzhou 510000, China

**Keywords:** *Staphylococcus aureus*, antimicrobial resistance, virulence

## Abstract

China is one of the largest producers of pigs and pork in the world. However, large-scale studies on pig-associated *Staphylococcus aureus* in relation to healthy pigs, diseased pigs and environment are scarce. The objective of the present study was to characterize and compare *S. aureus* isolates from healthy pigs, diseased pigs and environment through antimicrobial susceptibility testing, multiple locus sequence typing, *spa* typing, and antimicrobial resistance gene screening. Results showed all isolates were susceptible to linezolid and vancomycin. However, 66.7% (104/156) isolates were multidrug-resistant by displaying resistance to three or more antibiotics and high rates of resistance to penicillin, tetracycline, clindamycin, and clarithromycin were observed. Of the 20 multilocus sequence types (STs) identified among the isolates, ST9, ST188, and ST7 were most commonly isolated from healthy pigs and environment, while ST1 was most commonly isolated from diseased pigs. In total, 17 *spa* types were represented among the isolates, while t4792 was most commonly isolated from diseased pigs and t899, t189 were most commonly isolated from healthy pigs and environment. In conclusion, the genotypic and epidemiology characteristics observed among the isolates suggest pigs and pork could be important players in *S. aureus* dissemination.

## 1. Introduction

*Staphylococcus aureus* is an important opportunistic foodborne pathogen that can cause serious infections of the bloodstream, skin, and soft tissue in humans and animals [1]. According to reports, it can cause various suppurative infections and foodborne diseases in humans and animals, such as sepsis, pneumonia, mastitis, pericarditis, vomiting and diarrhea [2,3]. Human infections caused by pig-associated methicillin-resistant *Staphylococcus aureus* (MRSA) sequence type (ST) 398 indicate that pigs are a key reservoir of MRSA and these bacteria are transmitted to human through occupational contact with pigs [4]. Therefore, *S. aureus* is an important food safety hazard that requires special attention.

China is one of the largest pork producers in the world and houses more than 463 million pigs, accounting for 51.6% of all pigs worldwide [5]. In June 2015, the Chinese Academy of Sciences published a list of antibiotic use and final emissions in China and the report pointed out that of the 162,000 tons of antibiotics used in China in 2013, veterinary antibiotics accounted for 52%, while florfenicol, lincomycin, tylosin, and enrofloxacin are widely used in pig [6]. The frequent use of antibiotics and high feeding density on pig farms have facilitated the emergence and spread of *S. aureus* and livestock-associated methicillin-resistant *Staphylococcus aureus* (LA-MRSA) [7]. Antibiotics-resistant *S. aureus* have been isolated from pigs in farms, abattoirs, and markets in China. One study collected nasal swabs from 590 pigs in two abattoirs in Harbin and found 33.9% samples were *S. aureus*-positive, 38 samples were MRSA-positive, and ST398-t034 and ST9-t899 were most commonly isolated [7]. Another epidemiological survey of *S. aureus* in Danish retail meats found that the prevalence of *S. aureus* in turkey, pork and chicken was 86.96%, 75% and 78.43%, respectively [8], while ST398-t034 was the main type. Previous studies showed that raw and processed meat and food animals like pig as well as contaminated surfaces or tools could serve as vehicles for the transfer of *S. aureus* to foods in China [9,10], especially pig-associated ST9-t899-MRSA, which is the major strain [10].

In China, the prevalence of *S. aureus* has been paid attention to in different pig production chain, but there are few reports on the genotypic and epidemiology characteristics of *S. aureus* of the entire pork production chain. Therefore, the objective of this study was to characterize the prevalence of *S. aureus* colonization of healthy pigs, diseased pigs and environment and the antimicrobial resistance-associated phenotypes and genotypes and molecular characteristics of isolated *S. aureus*.

## 2. Results

### 2.1. Prevalence of S. aureus and MRSA

Among the 666 samples, 156 (23.4%) yielded *S. aureus* and 24 (3.6%) tested positive for MRSA (Table 1). Within the 156 *S. aureus* isolates, 121 (28.1%, 121/430) were from healthy pigs, 28 (17.0%, 28/165) were from diseased pigs, and 7 (9.9%, 7/71) were from environment. Among all types of samples, *S. aureus* had the highest prevalence in the abattoir (35.8%, 62/173), followed by the markets (23.1%, 37/160). MRSA prevalence ranged from 0% (0/97) from farm to 6.3% (10/160) from markets. Among the 28 *S. aureus* isolates of samples from diseased pigs, only three isolates (1.8%, 3/165) were MRSA.

### 2.2. Antimicrobial Susceptibility Profiles

All 128 *S. aureus* isolates, except those from diseased pigs, were susceptible to quinupristin-dalfopristin, linezolid, and vancomycin. Overall, as shown in Figure 1, isolates from pig farms, abattoirs, and markets were predominantly resistant to penicillin (98% to 100%), tetracycline (40–97%), and clindamycin (34–100%) and clarithromycin (40% to 93%). Many isolates were also resistant to gentamycin, ciprofloxacin, moxifloxacin), chloramphenicol, and other antibiotics. Resistance to a second-generation tetracycline, minocycline, was less prevalent than resistance to a first-generation tetracycline, tetracycline in general.

More than 90% of isolates from the farms were resistant to penicillin, gentamycin, tetracycline, clindamycin, and clarithromycin, as well as the fluoroquinolones ciprofloxacin and moxifloxacin (Figure 1). In addition, a large proportion of isolates from farms were resistant to levofloxacin (55.2%). higher resistance rates to the remaining antibiotics, with the exception of oxacillin, cefoxitin, chloramphenicol, minocycline and rifampin, were observed in farm isolates compared to the abattoir and market isolates. A high percentage of isolates from the abattoir were resistant to penicillin. Interestingly, the overall resistance prevalence was lower in abattoir isolates than market isolates for all tested antibiotics except gentamycin.

As indicated in Figure 1, isolates from diseased pigs displayed a high frequency of resistance to penicillin (100%), tetracycline (92.9%), clindamycin (96.4%), clarithromycin (96.4%), ciprofloxacin (96.4%), gentamycin (57.1%), and trimethoprim-sulfamethoxazole (42.9%). Resistance was found at low frequencies for cephalothin, oxacillin, cefoxitin, chloramphenicol, and rifampin. Notably, three isolates were resistant to quinupristin-dalfopristin.

As shown in Table A2 (Appendix A), only one *S. aureus* isolate was susceptible to all antimicrobial agents tested. The remaining isolates exhibited resistance to at least one of the antimicrobials. In total, 56 patterns of resistance for 9 categories of antimicrobials were found for these isolates, where the most common resistance pattern was PEN (28/156), followed by PEN-TET (11/156), PEN-GEN-TET-CLA-SXT-CLI-CIP-LEV-MXF (10/156), PEN-GEN-TET-CLA-SXT-CLI-CIP-MXF (9/156), PEN-GEN-TET-CLA-CLI-CIP (9/156), and Multidrug resistance was observed in 104 (66.7%) of the *S. aureus* isolates. Of the abattoir isolates, 30.6% (19/62) were multidrug-resistant (MDR), which is significantly fewer than farm isolates (22/22, 100%) and market isolates (32/37, 86.5%). Moreover, 85.7% (24/28) of the isolates from diseased pigs were MDR.

### 2.3. Prevalence of Resistance Genes

To evaluate the role of certain genes in the development of antimicrobial resistance, resistance-associated genes were amplified from the *S. aureus* isolates based on individual antibiotic resistance profiles. The results are presented in Table 2.

The gene *mecA* confers resistance to β-lactams and was present in 24 *S. aureus* isolates. Meanwhile, no *mecC* carriers were detected and *blaZ* was present in all isolates. The aminoglycoside resistance genes *ant*(4′)-*Ia*, *aadE*, *aacA*-*aphD*, and *aac*(6′)/*aph*(2”) were highly percentages among the isolates. Among the three genes associated with chloramphenicol resistance, *cmlA* was only detected in isolates from diseased pigs. A large proportion of isolates from diseased pigs carried the *tetM* gene, in contrast to isolates from other sources.

The *lsa*(E) gene, which is associated with pleuromutilin/lincosamide/streptogramin A resistance, has a high prevalence in farm isolates (100%, 22/22), abattoir isolates (93.5%, 58/62), market isolates (89.2%, 33/37) and diseased-pig isolates (96.4%, 27/28). Meanwhile, none of the isolates contained the multi-resistance gene variant of *cfr*(B). The *linA* gene, which confers resistance to clindamycin, was highly represented among the isolates from retail pork, while *lnu*(B) was detected at high frequencies in farm (100%, 22/22) and abattoir (62.9%, 39/62) isolates. Three genes associated with clarithromycin resistance, *ermA*, *ermB*, and *ermC*, were also detected. Only isolates from diseased pigs displayed a high prevalence of *ermA* and *ermB*, while *ermC* was detected in only three isolates, which were from farms and abattoir. The *dfrG* gene was frequently present, while *sulI*, *sulII*, and *dfrA* were rarely found.

### 2.4. Molecular Characteristics of S. aureus

Isolates were analyzed by multiple locus sequence typing (MLST) and *spa* typing (Figure 2). A total of 20 STs and 17 *spa*-types were identified. For farm isolates, the main MLST type was ST9 (77.3%, 17/22). And ST4292 was a newly discovered ST type in *S. aureus*. And there were 15 different ST types of isolates from the abattoirs, while the main types were ST188 (29.0%, 18/62), ST9 (21.0%, 13/62) and ST3387 (12.9%, 8/62). Meanwhile, nine different ST types have been identified in the market isolated, among them ST7 (40.5%, 15/37) and ST188 (18.9%, 7/37), were the main popular types.

Only one *spa* types, t899 (100%, 22/22) isolated from the farm. It can be seen that the types are relatively concentrated and single. In the environment, only one isolate with the *spa* type of t189 was observed and the others were all t899 type. The *spa* types of *S. aureus* isolated from the abattoirs were rich and diverse, including 13 types, mainly t189 (29.0%, 18/62) and t899 (32.3%, 20/62). There are seven *spa* types of *S. aureus* from the markets, mainly t091 (43.3%, 16/37) and t189 (24.3%, 9/37). The remaining types were t037, t899, t437, t1852 and t6675.

### 2.5. Genetic Relatedness

Based on the presence or absence of phenotypic resistance data and selected resistance genes, a dendrogram of similarity was established (Figure 2), which separated the 128 isolates in four main clusters. As determined by the dendrogram, the vast majority of *S. aureus* isolates clustered at the top of the dendrogram were abattoir isolates, and ten were market isolates. This cluster of isolates had diverse STs and *spa* types. In contrast, the remaining of the market isolates form two clusters towards the middle and the bottom of the dendrogram, respectively. Only seven abattoir isolates and the farm isolates clustered by dendrogram had identical STs and *spa* types, ST9 and t899, while the other abattoir isolates in this cluster had different STs and *spa* types.

## 3. Discussion

In order to monitor changes in the epidemiology and characteristics of *S. aureus*, some effective approaches like molecular typing were used in this study to analyze *S. aureus* isolates originating from healthy pigs, diseased pigs and environment. *S. aureus* was present in 9.9% to 35.8% of samples, where the most frequent contamination was observed in the abattoir (35.8%). MRSA prevalence ranged from 0% of samples from farms to 6.3% from markets. The overall prevalence of *S. aureus* and MRSA in our study was 23.4% and 3.6%, respectively. The average prevalence of *S. aureus* in this study was lower than that described in a study conducted in food and food animals in the Shaanxi province, China [11]. Similar results were obtained in a study in Hong Kong, where the prevalence of *S. aureus* in pigs was estimated to be 24.9% [12]. However, the results of our study appear higher (*p* ≤ 0.001) than another study for other pork samples or swine carcasses in the Shandong province, China (19.6%) [13]. Due to the complex environment of farms, markets and diseased pigs, *S. aureus* carrying rate of samples is high and relatively close, therefore we pay more attention to the genotype and phenotypic characteristics of strains from different sources.

Of the samples from the abattoir, 35.8% (62/173) were positive for *S. aureus*. The increased prevalence of *S. aureus* in samples after slaughtering compared to pigs on farms and retail meat suggest pig slaughtering process may play an important role in the transmission of *S. aureus*. During slaughtering, *S. aureus* can get transmitted via direct or indirect contact with the environment or meat products, including through processing machinery, refrigerators, meat containers, and staff. A study by Beneke estimated the occurrence of MRSA in abattoir environments was 12% in slaughterhouse samples [14]. One study also highlighted the role of the environment as a source of MRSA in the commercial pig production chain, where MRSA was detected in the pork production shower facilities of two commercial swine systems [15]. Another study found MRSA carriage by abattoir workers was caused by cross-contamination between workers and carcasses [16]. All of these elements acted as reservoirs and sources of pathogens and, consequently, lead to higher contamination levels. However, because the terminal samples did not all originate from the abattoir tested in this study, it was difficult to track contamination between transportation from the abattoir to the markets.

A total of 20 STs and 17 *spa* types were present in isolates from different sources. Abattoir isolates were notably diverse in MLST (*n* = 16) and *spa* type (*n* = 9) in, while market pork samples (*n* = 12 and 7, respectively) and farms (*n* = 3 and 1, respectively) displayed less diversity. LA-MRSA ST9 is currently the most prevalent sequence type in most Asian countries, but these strains are composed of different *spa* types, such as t899 in China, Hong Kong, and Taiwan [10,17,18], t4358 in Malaysia, and t337 in Thailand [19,20]. ST9-t899 was the most commonly detected type in our study and was especially prevalent on farms as it was identified in 17 and 9 isolates from the farms and abattoir, respectively, while most ST9 isolates are LA-MRSA, ST9 strains have been found to spread in both the presence and absence of direct livestock contact and even cause disease in humans [17,21].

Three major *S. aureus* sequence types ST188, ST9, and ST3387 were detected in the abattoir, while ST188-t189 accounted for 17.7% of the isolates and, similarly, was the most common lineage isolated from adults and adolescents with atopic dermatitis colonized by *S. aureus* in South Korea [22]. In another epidemiological study in Malaysia, ST188, ST7, and ST1 were all detected among the MRSA isolates from a public hospital [23]. ST7-t091 was the dominant type among isolates from the markets. ST7 has been found among MSSA isolates from slaughter pigs [7] and has been associated with skin and soft tissue infections in China [24]. While ST398 is the most prevalent LA-MRSA type in European countries and North America [25], its prevalence in our study was relatively low. ST239 is identified among hospital-acquired MRSA isolates [26], but was found in our market meat samples and in another study sampling pig farms [27]. Moreover, the ST5 lineage is believed to be the result of a human-to-poultry host jump followed by adaptation and then pandemic spread [28]. However, ST5 was isolated from slaughtered and diseased pigs in our study. In addition, the newly discovered ST types in pig source ST4292, ST4293, ST4294, ST4295, and ST4297 were confirmed to be present at the farm and abattoir stages. Therefore, these results indicate probable cross-contamination between humans and pork, indicating the need for more attention in further studies. 

The isolates from diseased pigs were mainly resistant to six to seven classes of antimicrobials with the common resistance pattern being PEN-GEN-TET-CLA-SXT-CLI-CIP. Notably, the diseased pig isolates had a high rate of resistance to penicillin (100%), tetracycline (92.9%), clindamycin (96.4%), clarithromycin (96.4%), and ciprofloxacin (96.4%). Of the isolates from diseased pigs, 85.7% were MDR, which may be the result of intensive and frequent exposure of the pathogens to antibiotics. This high prevalence of MDR isolates may lead to a reduction in the effectiveness of antimicrobial agents used in clinical treatments of humans, including for food-borne diseases. However, different farm sources may explain differences in MLST and *spa* types in isolates from healthy and diseased pigs. While the ST9-t899 clone was most frequently found in healthy pigs, ST1-t4792 was the most common one in diseased pigs. ST1 isolates are generally community-acquired MRSA [29] and also associated with staphylococcal food poisoning in South Korea and China [30,31]. Therefore, it is essential to continuously monitor the prevalence and resistance of *S. aureus* in livestock healthy and diseased pigs. Compared with isolates from other sources, isolates from diseased pigs have a higher gene carrying rate of *aadE*, *cmlA*, *tetM*, *ermA*, *ermB*, *sulI* and *sulII* (*p* ≤ 0.001) and *cmlA* was only detected in isolates from diseased pigs. However, we only focused on the characteristics of *S. aureus* in these samples in the current study. Therefore, we cannot draw a direct conclusion from our research that *S. aureus* causes clinical disease. To determine which pathogen caused the disease, high-throughput sequencing and clinical experiments are required, like histopathology immunohistochemistry [32] and metagenomics [33,34]. Future research should also elucidate the relationships among phylogenetic and clinical disease of *S. aureus* and expanding the sample size and sampling range is also necessary.

In our study, we found that *cmlA*, *sulII* and *dfrA* were only detected in diseased pig. The carrying rates were 32.1%, 42.9%, and 10.7%. However, *ermC* was only detected in healthy pigs. The most common detected gene combinations of the diseased pigs were *cmlA* + tetM + *sulI* (23/28), and all the strains that conform to this gene combination are ST1-t4792. However, due to the limited sample size, whether there is a direct link between typing and gene combination requires further study.

It is worth noting that the detection rate of *lsa*(E) gene in *S. aureus* isolated from this study was 94.2% (147/156), however in 2014, Yan et al. screened for *lsa*(E) gene from pigs’ isolates of two abattoirs in Harbin with a detection rate of only 22.0% (44/200). Among the strains isolated from the farm, the positive rate of the multidrug resistance gene cluster (*aadE*-*spc*-*lsa*(E)-*lnu*(B)-*tnp*) was 95.5%, and only one strain contained only the *lsa*(E) gene instead of the gene cluster. The multidrug resistance gene cluster is relatively conservative and is not susceptible to structural changes, which is an important reason for exacerbating the mutual transmission of resistance genes among different species.

## 4. Materials and Methods

### 4.1. Sampling

The 666 samples used in the present study were collected from the environment (*n* = 71), healthy pigs (*n* = 430) and diseased pigs (*n* = 165). All samples were collected in Fujian and Guangdong, China between December 2014 and June 2017.

Specifically, for the healthy pig samples, 173 samples were from abattoirs including 133 pork and 40 intestines, 160 were from markets and 97 were nasal swabs from 3 commercial swine farms. In addition, market samples were pork (*n* = 85), ribs (*n* = 24) and haslets (*n* = 51). As for environmental samples, there were pigpen gates (*n* = 18), soil (*n* = 20) and ground (*n* = 33) samples. Healthy pig samples and environmental samples are the same as our previous research [35]. All 165 samples were collected from pork (*n* = 60), brain (*n* = 23), and internal organs (*n* = 82) of diseased pigs. These samples of diseased pigs come from two different farms with symptomatic disease, including diarrhea and respiratory disease. The pork samples were about 250 g, and intestine samples were about 100 g. Sterile swabs were used to collect and preserve the environment and pig nose samples.

However, these terminal samples of meat from the markets did not all originate from the abattoir tested in the present study.

### 4.2. Bacterial Isolation and Identification

Isolation and identification of *S. aureus* were performed according to China’s National Technical Standard GB 4789.10-2016. Briefly, the samples were transferred to 7.5% NaCl broth and incubated at 37 °C for 24 h. The broth was then streaked onto Baird-Parker agar plates containing 5% potassium tellurite egg-yolk reagent (Huankai, Guangzhou, China) and incubated at 37 °C for 24–48 h. Presumptive *S. aureus* colonies were confirmed using plasma coagulase assays and were further screened by PCR amplification of the *nuc* gene [36]. One *S. aureus* isolate per sample was further analyzed. MRSA were determined by antibiotic phenotype and confirmed by PCR amplification of *mecA* using the primers listed in Table A1 (Appendix A).

### 4.3. Antimicrobial Susceptibility Testing

Antimicrobial susceptibility profile for each isolate was examined using broth microdilution for vancomycin and the disk diffusion method on Mueller-Hinton agar plates for the remaining antibiotics in accordance with current guidelines from the Clinical and Laboratory Standards Institute [37]. The following antimicrobial agents were tested, β-lactams antibiotics, penicillin (PEN, 10 μg), cefalotin (CEP, 30 μg), oxacillin (OXA, 1 μg), cefoxitin (FOX, 30 μg); aminoglycoside antibiotics, gentamicin (GEN, 10 μg); macrolide antibiotics, Clarithromycin (CLA, 15 μg); tetracycline antibiotics, tetracycline (TET, 30 μg), minocycline (MIN, 30 μg); lincosamides antibiotics, clindamycin (CLI, 2 μg); sulfonamides antibiotics, trimethoprim-sulfamethoxazole (SXT, 1.25/23.75 μg, respectively), ansamycin antibiotics, rifampin (RIF, 5 μg), fluoroquinolone antibiotics, ciprofloxacin (CIP, 5 μg), levofloxacin (LEV, 5 μg), moxifloxacin (MXF, 5 μg), gatifloxacin (GAT, 5 μg); oxazolidinone antibiotics, linezolid (LZD, 15 μg), and streptogramin antibiotics, quinupristin-dalfopristin (QDA, 15 μg). *S. aureus* strain ATCC 29,213 was used as a control. Isolates with intermediate levels of susceptibility were classified as resistant. MDR isolates were defined as having non-susceptibility to at least one agent in three or more antimicrobial categories.

### 4.4. Screening of Antimicrobial Resistance Genes

Bacterial genomic DNA was extracted from the isolates using a DNA extraction kit (Biomed, Beijing, China). The presence of potential antimicrobial resistance genes were selected according to the tested antimicrobials, and included *mecA*, *mecC*, *blaZ*, *aac*(6′)/*aph*(2″), *aph*(3′)-*IIIa*, *ant*(4′)-*Ia*, *aadE*, *tetK*, *tetL*, *tetM*, *ermA*, *ermB*, *emrC*, *msrA*, *msrB*, *catI*, *cmlA*, *fexA*, *dfrG*, *linA*, *lnu*(B), *lsa*(E), *cfr*(B), *sul**I*, *sul**II*. They were assessed by PCR and sequencing using primers listed in Table A1 (Appendix A) for all isolates.

### 4.5. Molecular Typing

All *S. aureus* isolates were characterized by multiple locus sequence typing (MLST) and *spa* typing, where fragments of seven housekeeping genes (*arcC*, *aroE*, *glpF*, *gmk*, *pta*, *tpi*, and *yqiL*) and the variable repeat region of the *spa* gene, were amplified by PCR and sequenced [38]. MLST were assigned using the default parameters listed on the MLST home page (http://www.mlst.net/) and *spa* type were assigned using the Ridom SeqSphere+ software [39].

### 4.6. Statistical Analyses

The dendrogram of similarity was established by using Jaccard’s coefficient of similarity and the unweighted-pair group method with arithmetic averages (UPMGA), which based on the presence or absence of phenotypic resistance data and selected resistance genes.

All statistical analyses were performed using SPSS 22.0 software. Rates of bacterial antimicrobial resistance were compared among different groups using the chi-squared test, where a *p* value of < 0.05 was considered statistically significant. 

## 5. Conclusions

In conclusion, 156 *S. aureus* isolated from 666 samples have been analyzed for genotypic and epidemiology characteristics. The multidrug resistance and diverse resistance patterns observed among the isolates suggest pigs and pork could be important players in *S. aureus* dissemination. In addition, *S. aureus* ST9-t899 was found colonizing pigs on the farms. Contamination of a downstream abattoir and retail market with these two types, respectively, revealed *S. aureus* clones were diverse across the pork production. Further studies are needed to examine additional risk factors for *S. aureus* colonization or infection that may be attributable to meat slaughtering and handling. This may indicate whether control strategies in the animal production process are feasible and could significantly reduce rates of *S. aureus* infections and carriage.

## Figures and Tables

**Figure 1 antibiotics-09-00839-f001:**
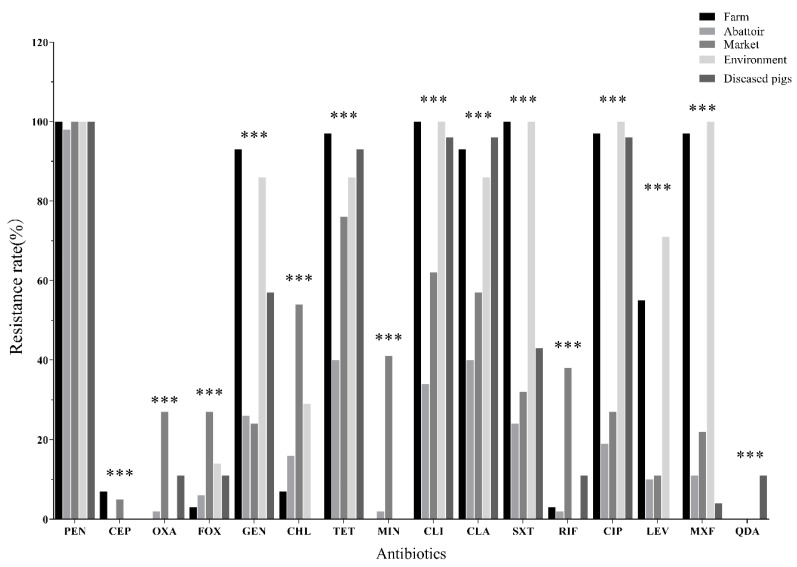
Occurrence of *S. aureus* isolate resistance to antimicrobials based on isolate source. *** *p* < 0.001; PEN, penicillin, CEP, cephalothin, OXA, oxacillin, FOX, efoxitin, GEN, gentamicin, CHL, chloramphenicol, TET, tetracycline, MIN, minocycline, CLI, clindamycin, CLA, clarithromycin, SXT, trimethoprim-sulfamethoxazole, RIF, rifampicin, CIP, ciprofloxacin, LEV, levofloxacin, MXF, moxifloxacin and QDA, quinupristin-dalfopristin.

**Figure 2 antibiotics-09-00839-f002:**
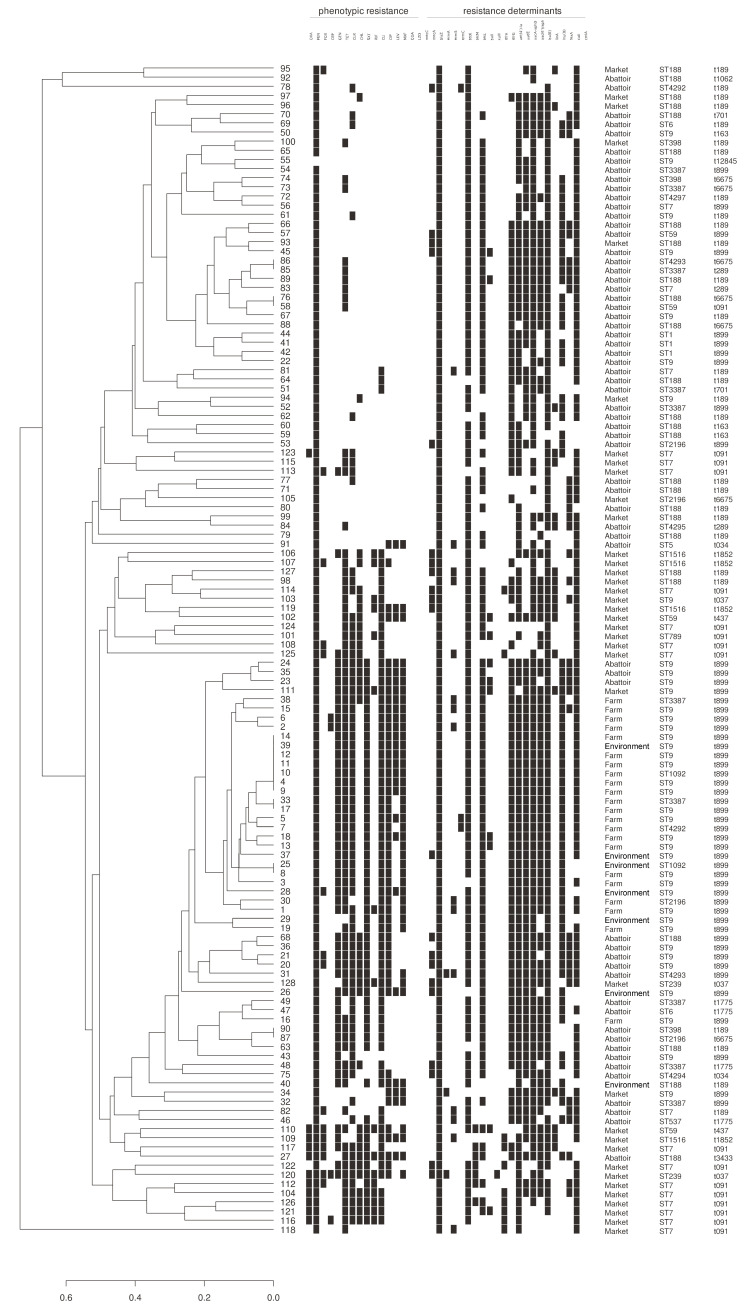
Dendrogram showing the resistance phenotypes and genotypes of *S. aureus* isolated from the pork production. The dendrogram was established based on the presence and absence of selected determinants or phenotypes using Jaccard’s coefficient of similarity and the unweighted-pair group method with arithmetic averages (UPMGA).

**Table 1 antibiotics-09-00839-t001:** Occurrence of *S. aureus* and MRSA in pork production.

Source		No. of Samples	No. (%) of Positive Samples	
			*S. aureus* Including MRSA ***	MRSA *
Healthy pigs	Farm	97	22 (22.7)	0 (0)
	Abattoir	173	62 (35.8)	9 (5.2)
	Market	160	37 (23.1)	10 (6.3)
Environment		71	7 (9.9)	2 (2.8)
Diseased pigs		165	28 (17.0)	3 (1.8)
All		666	156 (23.4)	24 (3.6)

Rates of positive samples were compared among different groups using the chi-squared test, * *p* < 0.05 and *** *p* < 0.001.

**Table 2 antibiotics-09-00839-t002:** Prevalence of antibiotic resistance associated genes among *S. aureus* isolates.

Antibiotics	Resistance Genes	No. (%) of Positive Isolates					
		Environment (*n* = 7)	Healthy Pigs (*n* = 121)			Disease Pigs (*n* = 28)	All (*n* = 156)
			Farm (*n* = 22)	Abattoir (*n* = 62)	Market (*n* = 37)		
Penicillin	*mecA*	2 (28.6)	0 (0)	9 (14.5)	10 (27.0)	3 (10.7)	24 (15.4)
	*blaZ*	7 (100)	22 (100)	62 (100)	37 (100)	28 (100)	156 (100)
Gentamycin	*ant*(4′)*-Ia*	7 (100)	22 (100)	53 (85.5)	28 (75.7)	9 (32.1)	119 (76.3)
	*aadE*	6 (85.7)	21 (95.5)	48 (77.4)	14 (37.8)	27 (96.4)	116 (74.4)
	*aacA-aphD*	7 (100)	22 (100)	58 (93.5)	28 (75.7)	28 (100)	143 (91.7)
	*aac*(6′)*/aph*(2”)	5 (71.4)	21 (95.5)	35 (56.5)	22 (59.5)	3 (10.7)	86 (55.1)
Chloramphenicol	*fexA*	0 (0)	1 (4.5)	25 (40.3)	9 (24.3)	4 (14.3)	39 (25.0)
	*cat* *I*	3 (42.9)	18 (81.8)	51 (82.3)	32 (86.5)	13 (46.4)	117 (75.0)
	*cmlA*	0 (0)	0 (0)	0(0)	0 (0)	9 (32.1)	9 (5.8)
Tetracycline	*tetK*	6 (85.7)	21 (95.5)	51 (82.3)	32 (86.5)	13 (46.4)	123 (78.8)
Tetracycline	*tetM*	0 (0)	0 (0)	1 (1.6)	6 (16.2)	19 (67.9)	26 (16.7)
	*tetL*	6 (85.7)	22 (100)	53 (85.5)	19 (51.4)	27 (96.4)	127 (81.4)
Clindamycin	*linA*	0 (0)	0 (0)	2 (3.2)	17 (45.9)	0 (0)	19 (12.2)
	*lnu*(B)	7 (100)	22 (100)	39 (62.9)	5 (13.5)	28 (100)	101 (64.7)
quinupristin-dalfopristin	*lsa*(E)	7 (100)	22 (100)	58(93.5)	33 (89.2)	27(96.4)	147 (94.2)
Clarithromycin	*ermA*	0 (0)	0 (0)	1 (1.6)	2 (5.4)	24 (85.7)	27 (17.3)
	*ermB*	1 (14.3)	4 (18.2)	5 (8.1)	5 (13.5)	24 (85.7)	39 (25.0)
	*ermC*	0 (0)	2 (9.1)	1 (1.6)	0 (0)	0 (0)	3 (1.9)
Trimethoprim-sulfamethoxazole	*sulI*	0(0)	2 (9.1)	4 (6.5)	5 (13.5)	25 (89.3)	36 (23.1)
	*sulII*	0 (0)	0 (0)	0 (0)	1 (2.7)	12 (42.9)	13 (8.3)
	*dfrA*	0 (0)	0 (0)	0 (0)	10 (27.0)	3 (10.7)	13 (8.3)
	*dfrG*	6 (85.7)	22 (100)	42 (67.7)	17 (45.9)	7 (25.0)	94 (60.3)

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
