# Peer review of "Genotypic Characteristics and Correlation of Epidemiology of Staphylococcus aureus in Healthy Pigs, Diseased Pigs, and Environment"

_antibiotics, 2020, doi:10.3390/antibiotics9120839_

Round 1

Reviewer 1 Report

General comments: The manuscript entitled “Genotypic characteristics and correlation of epidemiology of livestock-associated Staphylococcus aureus in healthy pigs, diseased pigs, and environment” describe S. aureus isolates from several sources.

Please indicate whether the isolates are the same ones that were used in a previous study “Prevalence of enterotoxin genes in Staphylococcus aureus isolates from pork production” and if so please clearly indicate this is the text and cite this manuscript then describe the differences of what was done in both studies.

Please add information regarding the diseased pigs and the source of the samples. Since the prevalence of S. aureus isolates was not higher in this group it is possible that the findings are not related to the clinical disease. Please discuss this point.

Please improve the introduction to better describe the relevant findings in pigs and specifically in China. I think that sentences such as "this study will serve as a model" should be removed or changed. 

Specific comments:

Title

I suggest deleting the words livestock-associated from the title since this is usually used to describe specifically MRSA and its use to describe S. aureus in general is confusing.

Abstract

I suggest changing or deleting the sentence: “this study will serve as model to investigate” in the conclusion. Instead, write a sentence that summarizes the specific findings.

Introduction

I suggest making a better distinction between S. aureus and MRSA.

I am not sure what is the significance of reference 7 from Denmark.

I think that better description and citation of studies, both from China and maybe other countries, regarding pigs and related products is required here.

“However, there is no report regarding the prevalence and characteristics of S. aureus isolated from healthy pigs, diseased pigs and environment in China” – this is not accurate and should be deleted or changed.

Results

Multi-drug or multidrug – it is written in both ways in the text (multi-drug in the abstract), please correct.

Page 2 – “MRSA prevalence ranged from 0% (0/165) from farm to 6.3% (10/160) from markets” – what is the number 165 refers to? did you mean farm or diseased pigs like in the sentence before?  

Page 2 – in 2.2 what is the number 128 isolates (121+7) refers to? Please add from healthy pigs and from the environment.

Fig 1. Please add that the isolates from farm (n=29) include isolates from the environment.

Page 8 – please write the full term MDR the first time it appears.

Page 8 – “were highly Percentages among” correct the capital letter.

Table 1. please add confidence intervals.

Table 1. please add a space after the number 22.

Table 2. I am not sure if this table is required and if so please suggest moving table 2 to the supplemental materials.

Table 3. Table 3. Correlation between phenotypic and genotypic resistance in S. aureus isolates. The word correlation should be corrected since the table describe the prevalence of the genes among the isolates and not the phenotypic resistance.

Page 9. Meanwhile, isolates from - please correct.

Page 9. Only one spa type, t899 (100%, 22/22) was isolated from the farm – please correct.

Page 9. There were 7 spa types – please correct.

Page 9. The remaining types were t037, t899, t437, t1852, and t6675. – please correct.

Fig 2. This is too small and phenotypic resistance and resistance determinants is not clear.

Discussion

“In order to monitor changes in the epidemiology and characteristics of S. aureus, a new approach was used…” – please change. I do not think that it is correct to write that this is a new approach.

First paragraph – “23.4and” add space after the number.

Please make sure that the difference is indeed statistically significant, please add confidence interval. “However, the results of our study appear higher than another study for other pork samples or swine carcasses in the Shandong, China (19.6ï¼…) [10] .”

Second paragraph – better comparison would be if the origin of the pigs in slaughterhouse and in the markets are all from the same farms.

Paragraph 3. – “. A total…” - delete the dot and the space in the beginning of the paragraph.

Page 11. “respectivelyWhile” – please correct.

Page 12. Was there a significant difference in resistance rates between samples collected from diseased pigs and other sources?

Materials and methods

Please write the timing (year) of sampling.

Please indicate if those are the same samples that were used for the study “Prevalence of enterotoxin genes in Staphylococcus aureus isolates from pork production”.

Please indicate the source of the clinical samples, nasal? Fecal? Both?

Please indicate whether the isolates from clinical cases was the cause.

The prevalence of S. aureus in the market, in the farms and in diseased pigs was similar and this should be discussed and maybe explained.

Conclusion

I suggest changing the sentence about different stages, since those were not directly compared.

References

Please make sure all are written in the same way, in some the full name of the journal is written whereas in other the abbreviation is written.

Author Response

Dear Editors,

We thank the reviewers for their comments on our manuscript entitled ‘Genotypic characteristics and correlation of epidemiology of Staphylococcus aureus in healthy pigs, diseased pigs, and environment’. Their input is very helpful and we have made significant changes and a very careful revision to improve the manuscript. Revised parts are marked in red in the manuscript. Please see the attachment

We believe that the manuscript is now suitable for publication in Antibiotics.

Dr. He Yan

School of Food Science and Engineering

South China University of Technology

On behalf of all authors.

Reviewer 2 Report

In the study some interesting results are presented but many parameters are reported and the number of samples per parameter is relatively small. As a result the experimentation is polyparametric and the comparison is not reliable. 

The number of samples from the environment is too small (7) so I suggest the grouping and comparison mainly for healthy and sick animals. Samples from the environment could be reported simply as an indicator.

Table 2 should be reorganized to be only one page.

More work (reorganization) should be done so as this paper to be acceptable for publication. 

Author Response

(The authors gave the same response as above.)

Reviewer 3 Report

This work investigates the carriage of S. aureus strains by Chinese pigs and studies the antibiotic resistance and clonal relationship of the isolates. This manuscript addresses an area of interest. However, some of the percentages reported in the text do not correspond with those in the figures and tables. I suggest a major review of all the data and calculations to make sure that the results reported are accurate.

Lines are not numbered to facilitate review

In this line I would use both times either the number of samples or the percentage of positive samples to be coherent :”One study collected nasal swabs from 590 pigs in two abattoirs in Harbin and found 33.9% samples were S. aureus-positive, 38 samples were MRSA-positive, and ST398-t034 and ST9-t899 were most commonly isolated”

Where do the samples come from? What regions of China?, is it all from the same location/farm? 24 samples is actually 3.6%.

(77.56%, 121/156) (17.95%, 28/156) (4.49%, 7/156). Among the 28 (17.9%, 28/156) S. aureus isolates of samples from diseased pigs, only three isolates (1.9%, 3/156) were MRSA. Review and correct all the percentages of the first paragraph of results.

According to table 1 there have been 156 S. aureus isolates, not 128.

What antibiotics are used in livestock in China? Include that information in the introduction.

Fig 1.- Statistically significant compared to what?

Why only 128 isolates have been studied and not the 156 isolated S. aureus? Fig 1

The percentages in the text do not correspond with the average percentages in figure 1. For instance, figure 1 indicates an average resistance to penicillin of >93%, almost 99%, and the text says 92.19%. Review and correct all percentages.

Table 2 is difficult to understand, for instance the two tables that appear in page 4 can be merged, and the tables that continue in page 5 and 6 need the lettering on top of the columns so they can be easier to read. I think the whole table can be merged to fit one page, or maybe two, but definitely not 4. With this table as it is currently displayed is difficult to know where the percentages in the text come from

Resistance was found at low frequencies for cephalothin, oxacillin, cefoxitin, chloramphenicol, and rifampin. Notably, three isolates were resistant to quinupristin-dalfopristin. Is this correlated with the antibiotics that are fed to the livestock?

Fix this in the text : ”and. ”

This:” a See the text for abbreviations. ” should be written at the end of the table and not in the middle.

There is a line in table 3 under mecA that should be removed

Table 3, indicate that the numbers in parenthesis are percentages

The two table 3 can be merged too.

The text says that “blaZ was not detected in any of isolates” but the table says the opposite, that it was found in all 156 isolates.

Table 3 the antibiotics do not align with the resistance genes, and ant(4’)-Ia seems to be a penicillin mechanism of resistance, please, fix the table so it is informative.

lsa(E), cfr(B) and mecC do not appear on the table

describe MLST on first appearance.

Sometimes the authors talk about 128 isolates, sometimes about 156 please fix if this is a mistake or clarify why there is a difference.

The quality of figure 2 makes impossible to read the dendogram in the upper part

Shaanxi province, Shandong province…

There is a paragraph in the discussion that starts with a “.” Remove. The same happens in the material and methods section

Respectively.

Why did the authors not study colistin resistance?

Author Response

(The authors gave the same response as above.)

Reviewer 4 Report

The topic raised by the authors of the manuscript titled "Genotypic characteristics and correlation of epidemiology of livestock-associated Staphylococcus aureus in healthy pigs, diseased pigs, and environment" was repeatedly discussed in the scientific literature. Although the manuscript is well written, there is no novelty in it. The data on antibiotic resistance of S. aureus are very broad, in the addition, the micro-dilution method is specified, and not the disc diffusion method, as in this manuscript. Moreover, although the collection of strains is large, they come from one geographic location which also significantly reduces the international aspect of the manuscript. It seems to me that the paper is more suitable for a more regional magazine. 

Author Response

(The authors gave the same response as above.)

Round 2

Reviewer 1 Report

Thank you, in my opinion the manuscript was improved and the authors did address all my comments.

please pay attention for typos and spaces like in line 55. 

Reviewer 2 Report

Dear authors

you tried to improve your work but there are thins should be improved. You can see my special comments in the attachment. 

Reviewer 4 Report

Thanks to the authors for responding to my earlier review